# Nitric Oxide Enhanced Salt Stress Tolerance in Tomato Seedlings, Involving Phytohormone Equilibrium and Photosynthesis

**DOI:** 10.3390/ijms23094539

**Published:** 2022-04-20

**Authors:** Lijuan Wei, Jing Zhang, Shouhui Wei, Dongliang Hu, Yayu Liu, Li Feng, Changxia Li, Nana Qi, Chunlei Wang, Weibiao Liao

**Affiliations:** College of Horticulture, Gansu Agricultural University, 1 Yinmen Village, Anning District, Lanzhou 730070, China; wlj920229@163.com (L.W.); zhkara@163.com (J.Z.); wsh920229@163.com (S.W.); h1468009331@163.com (D.H.); liuyayu199809@163.com (Y.L.); feng1654114573@163.com (L.F.); licxgsau5@163.com (C.L.); qnn2856108785@163.com (N.Q.); wangchunlei@gsau.edu.cn (C.W.)

**Keywords:** nitric oxide, RNA-Seq, salt stress, phytohormone, photosynthetic capacity, chlorophyll fluorescence

## Abstract

Nitric oxide (NO), as a ubiquitous gas signaling molecule, modulates various physiological and biochemical processes and stress responses in plants. In our study, the NO donor nitrosoglutathione (GSNO) significantly promoted tomato seedling growth under NaCl stress, whereas NO scavenger 2-(4-carboxyphenyl)-4, 4, 5, 5-tetramethylimidazoline-1-oxyl-3-oxide potassium (cPTIO) treatment reversed the positive effect of NO, indicating that NO plays an essential role in enhancing salt stress resistance. To explore the mechanism of NO-alleviated salt stress, the transcriptome of tomato leaves was analyzed. A total of 739 differentially expressed genes (DEGs) were identified and classified into different metabolic pathways, especially photosynthesis, plant hormone signal transduction, and carbon metabolism. Of these, approximately 16 and 9 DEGs involved in plant signal transduction and photosynthesis, respectively, were further studied. We found that GSNO increased the endogenous indoleacetic acid (IAA) and salicylic acid (SA) levels but decreased abscisic acid (ABA) and ethylene (ETH) levels under salt stress conditions. Additionally, GSNO induced increases in photosynthesis pigment content and chlorophyll fluorescence parameters under NaCl stress, thereby enhancing the photosynthetic capacity of tomato seedlings. Moreover, the effects of NO mentioned above were reversed by cPTIO. Together, the results of this study revealed that NO regulates the expression of genes related to phytohormone signal transduction and photosynthesis antenna proteins and, therefore, regulates endogenous hormonal equilibrium and enhances photosynthetic capacity, alleviating salt toxicity in tomato seedlings.

## 1. Introduction

Salt stress is one of the most detrimental environmental stressors and induces ionic toxicity, oxidative stress, and osmotic stress simultaneously [1]. Meanwhile, salinity destroys the processes of plant growth and development by inhibiting the ability to take up water and nutrients. In addition, K^+^ retention and Na^+^ exclusion are the key factors involved in salt resistance [2]. Salt stress is usually accompanied by oxidative damage due to excess production of reactive oxygen species, causing the oxidation of protein, membrane lipids, and nucleic acids, thereby inhibiting plant growth and development [3]. Additionally, plant physiological physiognomies are extensively susceptible to high salt toxicity. Salinity leads to changes in plant growth and development at the physiological and molecular levels [4]. It interferes with photosynthetic activity and respiration; impairs ion homeostasis and osmotic and hormonal balance; and reduces enzyme activity, protein and nucleic acid synthesis, and organic solute accumulation, thereby inhibiting seed germination, seedling growth, plant life, and crop productivity [2,5]. As a result, salt stress leads to various physiological and molecular changes and impedes plant growth by impairing photosynthesis, thereby decreasing the available resources and repressing cell division and expansion [6]. Consequently, the enhancement of resistance to salinity in crop plants is currently an urgent requirement.

Recently, the signaling of gases, such as hydrogen sulfide, nitric oxide (NO), carbon monoxide, and hydrogen, has been shown to play a vitally important role in various physiological processes in plants, [7,8,9]. Among these gases, NO, a small ubiquitous signaling molecule, plays important roles in plant growth and development processes, including seed germination, root development, seedling growth, fruit ripening, and postharvest preservation [10,11,12]. NO is produced mainly via two pathways in plants: non-enzymatic and enzymatic pathways [11]. Moreover, a study indicated that NO can interact with other molecules to modulate plant growth processes and respond to abiotic stress [13]. Moreover, the involvement of NO in heavy metal stress [11], drought stress [14], chilling stress [15], osmotic stress [16], and salt stress [17] has been elucidated. Therefore, the roles of NO in plant growth, development, and abiotic stress responses is a topic of great interest.

Previous studies have shown that NO treatment can effectively delay the softening and ripening of papaya fruit, likely via the regulation of certain endogenous hormones [18]. In various model plants or crop species, the roles of hormones in regulating the process of plant growth, development, and responses to stress stimuli have extensively been reported. Various important genes, transcription factors, and molecular players have been indicated to have essential effects on the mediation of multiple developmental processes and stress responses modulated by hormones [19]. Interestingly, there is now compelling experimental evidence that NO usually mediates various signaling processes through interactions with different phytohormones to affect heavy metal tolerance, notably via the post-translational modification of protein *S*-nitrosylation in response to heavy metal stress [20]. These findings imply a relationship between NO and phytohormones in responding to abiotic stress in plants. In plant leaves, red and blue-violet light are the main absorption spectra of photosynthetic pigments. Additionally, photosynthesis mainly requires red light, and blue light is important for plants in regulating various physiological processes, such as chlorophyll synthesis, chloroplast development, and stomatal development [21,22]. Photosynthesis, together with cell growth, is among the primary processes affected by salinity. The effects are direct, such as decreased carbon dioxide availability caused by limitations in diffusion through the stomata and the mesophyll or changes in photosynthetic metabolism. In addition, they can arise as secondary effects, namely oxidative stress, seriously affecting leaf photosynthetic machinery [23]. Thus, in this study, RNA sequencing (RNA-seq) technology was used to identify the candidate genes associated with plant hormone signal transduction and photosynthesis during NO alleviation of salt stress in tomato seedlings. This study may further improve our understanding of NO signal transduction in molecular mechanisms and provide initial insights into the regulatory role of NO in response to salt stress.

## 2. Results

### 2.1. The Effect of NO on the Growth of Tomato Seedlings under Salt Stress

To explore the effect of NO on the growth and development of tomato seedlings under salt stress, the NO donor GSNO and the NO scavenger cPTIO were used to treat tomato seedlings under salt stress. As shown in Figure 1, NaCl treatment significantly inhibited the growth of tomato seedlings compared with the control, whereas GSNO significantly reversed the inhibitive effect of NaCl. Compared with NaCl treatment alone, treatment with GSNO significantly increased the leaf area, root length, plant height, dry weight, fresh weight, and root activity of tomato seedlings by 27.6%, 25.3%, 23.0%, 44.8%, 22.3%, and 40.1%, respectively (Figure 1). Under NaCl stress, the tomato leaves became yellow and small, and the roots became less active and shorter, whereas GSNO alleviated the variation in phenotypes (Figure 1e,f). Interestingly, the NO scavenger cPTIO decreased leaf area, root length, plant height, dry weight, fresh weight, and root activity under salt conditions in comparison with NaCl + GSNO treatment. Interestingly, cPTIO treatment alone had a slightly inhibitory effect on tomato seedlings compared with the control.

### 2.2. Identification and Functional Classification of DEGs during NO Alleviation of Salt Stress

To further understand the basis of the molecular mechanisms of NO on the alleviation of salt stress in tomato seedlings, RNA-seq analyses were conducted for the different treatments. As shown in Appendix A, RNA-seq yielded 45–65 million raw reads. We discarded low-quality reads, which contained adapters and unknown or low-quality bases. After stringent quality checks and data cleaning, more than 44.58 million clean reads and 48.63 G clean bases were obtained. The average percentages of Q20 and Q30 reached 97.73 and 93.47%, respectively. The GC (guanine + cytosine) contents of these samples were about 42%, and the GC content of each group remained stable and basically unchanged, indicating a higher quality of sequencing that could be used for further analysis. As shown in Figure 2, in NaCl + GSNO vs. NaCl, according to the principle of “*p*-value less than 0.05”, a total of 739 DEGs were obtained through RNA-seq (Figure 2a, Appendix A). Among these DEGs, 313 genes were upregulated and 426 were downregulated, and these were significantly regulated by GSNO treatment, implying that these DEGs might be associated with various metabolic pathways. Moreover, these identified DEGs were significantly annotated with 10 biological processes, 5 cellular components, and 10 molecular functions in GO categories (Figure 2b, Appendix A). The terms of regulation of “cell wall organization or biogenesis (12),” “anion transport (10),” “external encapsulating structure organization (9),” “cell wall organization (9),” and “cell wall modification (7)” were mainly enriched in the biological processes. In molecular function, “enzymes inhibitor activity (18),” “enzymes regulator activity (18),” and “molecular function regulator (19)” were the dominant groups. According to the comparison against the KEGG database, in our results, a total of 292 enriched genes were assigned to 86 KEGG pathways (Appendix A). Additionally, the top 20 KEGG pathways corresponded to the most abundant DEGs, as shown in Figure 2c, which indicated that a high percentage of obviously annotated DEGs was enriched in the processes of photosynthesis antenna proteins, plant hormone signal transduction, and carbon metabolism and the MAPK signaling pathway. Therefore, the results mentioned above imply that NO may play an important role in regulating these pathways during NO alleviation of salt toxicity. Among these pathways, the plant hormone signal transduction pathway and photosynthesis antenna proteins were selected for subsequent analysis.

### 2.3. Analysis and Confirmation of DEGs Involved in Plant Hormone Signal Transduction during NO Alleviation of Salt Stress

In NaCl + GSNO vs. NaCl, 16 DEGs were identified that contributed to the hormone signal transduction pathway (sly04075) (Figure 3 and Appendix A). In addition, the hormone signal transduction pathways included the indoleacetic acid (IAA), abscisic acid (ABA), ethylene (ETH), salicylic acid (SA), gibberellin (GA), and cytokinin (CTK) signaling pathways. There were 5 DEGs identified in the AUX signaling pathway; auxin/indoleacetic acid (AUX/IAA), auxin-responsive protein (SAUR), and GH3 are the key genes involved in the auxin signal transduction pathway. *IAA35* was significantly downregulated in the NaCl + GSNO treatment in comparison with NaCl stress. The SAUR gene SAUR50 was downregulated in the NaCl + GSNO treatment compared with NaCl treatment alone. In addition, three GH3 genes (*CH3.1*, *CH3-4*, and *CH3-9*), which induce IAA production, were upregulated by GSNO (Figure 3 and Appendix A). In the ABA signaling pathway, the expression levels of the serine/threonine-protein kinase gene (*SRK2I*) were significantly downregulated by GSNO. Meanwhile, of the two *PP2C* (protein phosphatase 2C) genes, *PP2C 51-like* was downregulated, while *PP2C 75* was upregulated in the presence of GSNO. The EBF (EIN3-binding F-box protein) genes *EBF2* and *EBF1* as well as the ERF (ethylene response factor) gene *ERF-C3* were downregulated after adding GSNO under salt stress. Additionally, some DEGs involved in the SA, GA, and CTK signaling pathways were expressed in upregulated patterns (*LOC101265854*) or downregulated patterns (*PR1*, *EIL1*, *LOC101265384*, and *ORR26*) (Figure 3 and Appendix A).

To determine the reliability of our transcriptome data, the expression of 9 DEGs related to plant hormone transduction during NO alleviation of salt stress was conducted (Figure 4). These DEGs included three AUX-related genes (*GH3.1*, *IAA35*, and *GH3-9*), two ABA-related genes (*SRK2I* and *PP2C 51-like*), two ETH-related genes (*ERF-C3* and *EBF2*), and two SA-related genes (*LOC101265854* and *EIL1*). Compared with NaCl, the expression levels of *GH3.1*, *GH3-9*, and *LOC101265854* exhibited a significantly increased trend in the NaCl + GSNO treatment, while the expression levels of *IAA35*, *SRK2I*, *PP2C 51-like*, *ERF-C3*, *EBF2*, and *EIL1* were significantly decreased in the NaCl + GSNO treatment. These results show that all the expression patterns analyzed by qRT-PCR were consistent with our transcriptome analysis, confirming the reliability of the RNA-seq data.

### 2.4. Analysis and Confirmation of DEGs Involved in the Photosynthesis during NO Alleviation of Salt Stress

According to the KEGG results, nine significantly enriched DEGs in the photosynthesis antenna proteins pathway (sly00196) (Figure 5 and Appendix A) were involved in the salt stress response. Among the nine DEGs, six genes were annotated as light-harvesting chlorophyll complexes of photosystem II (LHCII) of photosystem II (PSII), including genes encoding light-harvesting complex II, chlorophyll a/b binding protein 1 LHcb1 (*chlorophyll a-b binding protein 1C* (*Cab-1C*), *Cab-3C*, *Cab-1A*, and *Cab-1D*), LHcb3 (*CAB13*), and LHcb2 (*CAB5*). In addition, these six DEGs annotated as LHCIIs of PSII were significantly upregulated by GSNO + NaCl treatment compared with NaCl treatment alone. Furthermore, three DEGs, light-harvesting complex I, chlorophyll a/b binding protein 1 Lhca1 (*LOC101252151*), and LHca4 (*CAB11* and *CAB12*), enriched in light-harvesting chlorophyll complexes of photosystem I (LHCI), were significantly upregulated after adding GSNO under stress.

To verify the expression of DEGs, four photosynthesis-related genes were selected. As shown in Figure 6, the expression levels of *CAB13*, *Cab-3C*, *LOC101252151*, and *CAB11* were decreased under NaCl stress compared with the control. Meanwhile, the expression of these genes under NaCl + GSNO treatment increased, showing about a one-fold increase in comparison with NaCl treatment alone. However, this increase was inhibited by cPTIO under salt stress compared with NaCl + GSNO treatment. In addition, cPTIO treatment alone also slightly decreased the expression level of these photosynthesis-related genes, implying that NO probably alleviated salt stress by regulating photosynthesis.

### 2.5. Changes in the Contents of Endogenous Hormones during NO Alleviation of Salt Stress

Since 16 DEGs were related to “hormone signal transduction”, most of the DEGs were involved in the IAA, ABA, ETH, and SA signal transduction pathways. In order to determine whether NO could also affect plant endogenous hormones in response to salt stress, the content changes of four plant hormones in tomato seedlings, including IAA (the most common auxin), ABA, ETH, and SA, were determined (Figure 7). In comparison with the control, NaCl treatment significantly reduced IAA and SA contents while increasing ABA and ETH contents. However, GSNO treatment significantly reversed the effect of NaCl. Additionally, NaCl + cPTIO treatment decreased IAA and SA contents by 28.82% and 24.56%, respectively, in comparison with NaCl + GSNO treatment. Meanwhile, ABA and ETH content was decreased by NaCl + GSNO treatment compared with NaCl treatment alone, which was reversed after adding cPTIO under salt stress. However, for cPTIO treatment alone, the hormone contents were significantly different compared with the control, except for SA content. Thus, these results indicate that NO significantly increased the levels of IAA and SA and reduced the contents of ABA and ETH to alleviate salt toxicity in tomato seedlings.

### 2.6. Changes in Photosynthesis Pigments and Chlorophyll Fluorescence Parameters during NO Alleviation of Salt Stress

To explain the stimulatory roles of NO in the alleviation of salt stress involving photosynthesis, the photosynthesis pigments and chlorophyll fluorescence parameters in tomato seedlings were determined in this study (Figure 8). As shown in Figure 8, NaCl treatment significantly decreased the contents of chlorophyll and carotenoid compared with the control. However, compared with NaCl treatment alone, NaCl + GSNO treatment increased the contents of chlorophyll a, chlorophyll b, and carotenoid. Additionally, when treated with NaCl + cPTIO, this increase was inhibited in comparison with NaCl + GSNO treatment (Figure 8a,b). Compared with the control, cPTIO treatment alone had no obvious effect on the content of carotenoid, whereas it significantly reduced the chlorophyll content (Figure 8a,b). GSNO treatment had a significant effect on the chlorophyll fluorescence parameters of tomato leaves under salt stress (Figure 8c–f). The maximum photochemical efficiency (Fv/Fm), actual photochemical efficiency (φPS II), and the photochemical quenching (qP) in the NaCl treatment were significantly lower than those in the control (Figure 5), whereas they were further increased after adding GSNO. In addition, when treated with NaCl + cPTIO, this increase was inhibited in comparison with NaCl + GSNO treatment. The Fv/Fm and φPS II of the cPTIO treatment alone showed a lower trend compared with those of the control. The non-photochemical quenching (NPQ) in the NaCl + GSNO treatment was significantly lower than that in the NaCl treatment, and the NPQ was smaller with GSNO supplementation. Thus, NO could effectively increase photosynthesis pigment content and then enhance photosynthetic capacity under salt stress.

## 3. Discussion

NO, a ubiquitous bioactive gas, plays an important role in plant growth and development and the adaptation of plants to stress conditions. Increasing evidence shows that the supplementation of exogenous signaling molecules can enhance the resistance of plants to salt toxicity by stimulating biochemical and physiological defense systems in plants [8,11,24]. NO, a multifunctional messenger molecule, functions in the amelioration of various types of abiotic stress, including salt stress [4]. The results presented herein indicate that NO can increase the leaf area, root length, plant height, dry weight, fresh weight, and root activity of tomato seedlings under NaCl stress, thereby alleviating salt toxicity and promoting the growth of tomato seedlings under salt stress (Figure 1). Moreover, GSNO treatment also alleviated the phenomenon of leaf yellowing compared with NaCl treatment alone (Figure 1). Thus, these results indicate that NO is responsible for enhancing the tolerance of tomato seedlings to salt stress. Piacentini et al. [25] revealed that increased endogenous NO could alleviate the inhibition of adventitious root elongation and lateral root formation and decrease the increment in lignin deposition induced by Cd in rice, thereby preventing root system alterations. Furthermore, NO ameliorated salt toxicity by modulating the physiological responses in seed germination and the early seedling growth characteristics of pak choi [26]. In addition, NO has been found to govern every single step of plant growth by balancing antioxidants and ROS under adverse conditions, which may be one of the important reasons why NO alleviates salt stress [1,3]. Interestingly, NO itself is considered a reactive nitrogen species, and it has been shown to be a potent antioxidant or a potent oxidant in plants [27]. However, higher-concentration GSNO (200 μM) treatment had no significant effect on the growth of plants [4,28]. Thus, although lower concentrations of NO can exert an effective effect in response to stress stimuli, high concentrations of NO might be toxic to plants because of its high reactivity, revealing the concentration-dependent effect of NO on the alleviation of environmental toxicity. To further determine the effect of NO on tomato seedling growth under salt stress, the NO scavenger cPTIO was applied in our study. Our results found that NaCl + cPTIO treatment significantly inhibited the growth of seedlings compared with NaCl + GSNO (Figure 1). In addition, after cPTIO treatment alone, the growth of tomato seedlings was slightly inhibited compared with the control. Similarly, Arora and Bhatla [29] found that NO promoted the growth of sunflower seedlings under salt stress, which was worsened after supplying cPTIO, implying that endogenous NO scavenging can prevent plant growth. Thus, these results indicate that NO may play an indispensable role in reducing salt-induced damage to seedling growth.

The major hormones in plants are auxins (AUXs), cytokinins (CTKs), abscisic acids (ABAs), gibberellins (GAs), ethylene (ETH), salicylic acid (SA), jasmonates (JAs), brassinosteroids (BRs), and strigolactones. Among them, ABA, JA, SA, and ETH are indicated to play major roles in the response to and plant defense against abiotic stress [30,31]. Generally, ABA is especially responsible for plant defense response against abiotic stress, as unfavorable conditions, including drought, cold, salinity, heat stress, and wounding, can trigger an increase in endogenous ABA levels [32,33]. Recently, AUXs were also shown to play an integral role in plant responses under adverse environmental conditions. In our study, NO significantly increased the levels of IAA and SA and decreased the levels of ABA and ETH during salt stress. Similarly, NO may alleviate Al stress by increasing IAA levels in the root apices to enhance Al tolerance in the root apices of rye and wheat [34]. These observations could account for the change in gene expression levels involved in the hormone signal transduction in the NaCl + GSNO treatment. Many key genes involved in various AUX-related signal pathways are invariably expressed differentially under different stress conditions [35,36]. In addition, these auxin-responsive genes have been broadly grouped into three major classes: Aux/IAA, SAUR, and GH3 [37]. In our study, *CH3.1*, *CH3-4*, and *CH3-9* genes were upregulated, and *IAA35* and SAUR50 were downregulated by NO under salt stress, implying that NO might mediate the degradation of Aux/IAA repressors to induce auxin-regulated responses. GH3 genes can maintain endogenous AUX homeostasis by conjugating excess AUX with amino acids. Singh et al. [38] indicated that GH3 family genes were highly induced under drought and/or salt stress, suggesting a role of GH3 family genes in abiotic stress responses. However, a previous study indicated that *SAUR75* was downregulated in wheat roots under salt stress, and the overexpression of *SAUR75* increased drought and salt tolerance in Arabidopsis [39]. These results indicated that NO-increased IAA levels induced the dynamic balance of AUX-responsive gene expression, thereby enhancing salt tolerance in plants. Furthermore, the significance of SA has been increasingly recognized in improved plant stress tolerance through SA-mediated plant metabolic processes. Alternatively, NO might be involved in SA-induced reduction of oxidative damage to confer salt tolerance in rice seedlings [40], implying that the relationship between NO and SA is in response to salt stress. EIL1 is known as a repressor of SID2 (SALICYLIC ACID INDUCTION DEFICIENT 2), a gene encoding an isochorismate synthase required for SA biosynthesis [41], which further explains the NO-induced upregulation of ELI1 expression and the increase in SA content under salt stress in this work (Figure 3, Figure 4 and Figure 7). The core ABA signaling is composed of PYR/PYL/RCAR receptors, PP2C phosphatases, and SnRK2 kinases. We found that NO further reduced the ABA content under salt stress, upregulated *PP2C 75,* and downregulated *PP2C 51-like* and the serine/threonine-protein kinase gene *SRK2I* (Figure 3, Figure 4 and Figure 7). This may be because the ABA-bound PYLs form complexes with the clade A PP2C, which allows the release of the inhibition of the serine/threonine-protein kinase gene SnRK2 protein kinases by PP2C [42]. However, Zhang et al. [43] suggested that NO upregulated the content of ABA and then enhanced water stress tolerance in maize leaves. Plants accumulate ABA quickly and in turn activate different stress responses when subjected to abiotic stress. When environments are optimal, the ABA level is decreased to a basal level, which enhances optimal growth. The regulation of ABA levels in tissues and cells is critical for balancing defense and growth processes when plants face non-optimal environments [44]. Accordingly, it could be speculated that NO-induced changes in key gene expression in ABA signaling inhibited ABA signaling, which resulted in a reduction in the ABA level to a basal level and thus alleviated salt toxicity. ETH serves as a small gas signal molecule in plant growth and development, such as adventitious root formation, hair formation, seed germination, flower and leaf senescence, and fruit ripening. Moreover, recently, ETH was shown to play a remarkable role in regulating the networks of plant signaling, which responded to multiple types of biotic and abiotic stress [45]. NO was also reported to impair ETH biosynthesis, thereby improving the vase life and quality of cut rose flowers [10], which was consistent with our results (Figure 7). Interestingly, in Mg-deficient Arabidopsis, ETH could facilitate NO production [46], revealing a novel signaling pathway of NO and ETH in response to abiotic stress. Thus, these results suggest that NO may alleviate salt damage in tomato seedlings by regulating hormonal levels. Interestingly, GSNO-induced regulation of these genes’ expression under stress was reversed by adding the NO scavenger cPTIO (Figure 4), further confirming the necessary roles of NO in regulating plant hormone signal transduction under salt stress.

It is well known that photosynthesis in plants begins with the absorption of light energy by photosynthetic pigments, and the photosynthetic capacity of plants can be indirectly reflected by the content of photosynthetic pigment in plant leaves [4]. Additionally, chlorophylls can capture light energy from the sun to dominate the photosynthetic capacity of plants. Thus, chlorophylls are one of the most essential photosynthetic pigments [47]. However, adverse conditions decrease the chlorophyll content. Previous studies have reported increased chlorophyll accumulation with increasing stress levels in stress-tolerant plants [47]. In our study, nine DEGs (LHC family) enriched in the photosynthesis antenna protein pathway were upregulated by NO under salt stress (Figure 4 and Figure 5), indicating the important role of NO in enhancing the photosystem II capacity under salt stress. It was reported that the downregulation of any member of the LHCB family reduced responsiveness of stomatal movement to ABA and then led to a decrease in plant tolerance to drought stress in Arabidopsis [48]. Meanwhile, the contents of chlorophyll a, chlorophyll b, and carotenoids in cucumber leaves were significantly increased by GSNO treatment (Figure 8). In addition, the measurements of chlorophyll level and leaf area are an important proxy for crop productivity. A previous report indicated that NO significantly delayed the senescence of cotton leaves and increased the chlorophyll content under salt stress [49]. In cucumber leaves, light supplementation enhanced the photosynthetic capacity by increasing the contents of chlorophylls and carotenoids [22]. Similar results were also found in our work, implying the importance of chlorophylls and carotenoids in plant growth and stress response. That is to say, the level of photosynthetic pigment was an essential indicator in reflecting the photosynthetic capacity of plants under adverse conditions. Therefore, the interrelationship between stress conditions and photosynthetic pigments might be a potential and important line of inquiry in the future. PSII is generally considered to be the most sensitive part of photochemistry under various abiotic stress. In our results, by chlorophyll fluorescence parameter analysis, we found that salt stress caused a significant decrease in Fv/Fm, actual φPS II, and qP in tomato leaves, which was accompanied by similar decreases in the coefficient of NPQ (Figure 8). This implies that the reduced efficiency of primary light energy conversion and the energy capture of these open centers leads to a decrease in PSII activity [50]. However, NO improved Fv/Fm, actual φPS II, and qP and reduced the coefficient of NPQ (Figure 8). These results indicate that NO might alleviate salt stress in tomato seedlings by activating photosystems. This is consistent with research regarding NO in bamboo [50], which showed that NO protected PSII activity from the acid rain stress by accelerating the electron transport activity and photochemical efficiency, which protected the photosynthetic apparatus from acid rain toxicity. That is to say, NO effectively improved the electron transport activity and photochemical efficiency, thus stimulating PSII activity under salt stress. In our results, NO decreased NPQ under salt stress, which resulted in a decrease in the non-photochemical loss of absorbed excitation energy and reduced the possibility for the efficient use of absorbed light for photochemistry, which might be due to NO-induced structural reorganizations and antenna complexes of PSII [51]. Thus, NO effectively improved the photosynthetic pigment content and chlorophyll fluorescence parameter and promoted PSII activity, thereby alleviating salt toxicity in tomato seedlings.

## 4. Materials and Methods

### 4.1. Plant Materials and Treatment

The tomato “Micro-Tom” was used as the material in our study. The seeds were disinfected with 1% of NaClO and transferred to 1/2 Hoagland solution for 7 days after germination, then continued cultivation in Hoagland solution for another 21 days. The uniform seedlings were chosen for treatment with control (distilled water), NaCl (150 mM), NaCl + nitrosoglutathione GSNO (10 μM) (a NO donor), NaCl + NO scavenger 2-(4-carboxyphenyl)-4, 4, 5, 5-tetramethylimidazoline-1-oxyl-3-oxide potassium (cPTIO) (10 μM), and cPTIO after adding Hoagland solution for 7 days. The concentrations of NaCl, GSNO, and cPTIO were based on the results of a preliminary experiment. The experimental environment was kept at 16 h light (250 μmol m^−2^ s^−1^ photon irradiance) at 26 °C and 8 h dark at 20 °C in 60% relative humidity.

### 4.2. Measurement of Morphological Indexes

Leaf area was measured by a leaf area scanner (YMJ-C, Zhejiang Topp Co., LTD., Taizhou, China) on whole single plants. Plant height was obtained by a Vernier caliper using the straight-line distance from the stem base to the apex of the shoot apical meristem. After removing the aboveground part of the seedlings, the image of the roots was scanned by a root scanner (STD4800, Quebec, QC, Canada), and then the total root length of each plant was obtained by the root analysis software WinRHIZO 5.0 (Regent Instruments, Inc., Quebec, QC, Canada). The mean values of plant height, leaf area, and total root length were measured for five tomato seedlings in each replication.

### 4.3. Measurement of Physiological Indexes

The fresh weight (FW) and dry weight (DW) were determined using the aboveground part of the tomato seedlings. The FW was measured using an electronic balance (SQP, Sartorius, Shanghai, China), and after the plant samples were dried at 80 °C for 48 h, which was when they reached a constant weight, the DW was measured. Then, the root activity was measured by the triphenyl tetrazolium chloride (TTC) method as described by Yan et al. [32] with minor modification. The tomato roots (about 1 g) were washed with deionized water and excised at 2 cm from the root tips at 2 h with a mixture of 7 mL 0.4% (*w*/*v*) TTC and 7 mL phosphate buffer (pH 7.5) at 37 °C, and then 4 mL of H_2_ SO_4_ (1 M) was added. Roots were dried with filter paper and extracted with ethyl acetate. The absorbance was measured at 485 nm.

### 4.4. RNA Extraction and Transcriptome Sequencing Analysis

Plant samples treated with 150 mM NaCl and 150 mM NaCl + 10 μM GSNO were used for RNA transcriptome sequencing analysis. Total RNA was extracted using the TaKaRa MiniBEST plant RNA extraction kit (Takara, 9769, Takara Bio Inc., Kusatsu, Shiga, Japan), and more than 1 μg of RNA samples were collected for RNA-seq library construction according to the instructions of the Illumina Standard mRNAseq library preparation kit (Illumina, CA, USA). The RNA-seq libraries were sequenced on the Illumina HiSeq platform to generate 150 nt paired-end reads. Then, raw data were filtered with Cutadapt software, and clean reads were mapped to the reference genome sequence of tomato SL2.40 (ITAG2.3) by use of TopHat. The values of fragments per kilobase of transcript per million reads for each biological replicate were determined using Cufflinks. The expression level of each gene was normalized by the value of fragments per kilobase of transcript per million fragments mapped (FPKM). The DEGs between different treatments were employed and analyzed using the DESeqR package. DEGs were defined on the basis of an adjusted absolute log 2 (fold change) > 0 and a *p*-value < 0.05. GO enrichment and KEGG pathway annotation were detected using BLAST2GO software and KOBAS software, respectively [52,53].

### 4.5. Quantitative Real-Time PCR (qRT-PCR) Assays

To validate the RNA-seq results, 9 DEGs involved in the phytohormone signal transduction pathway and 4 DEGs involved in photosynthesis antenna proteins were selected, and qRT-PCR assays were performed. Total RNA extraction, cDNA synthesis, and qRT-PCR were conducted as described by Gao et al. [54]. All the primers used for RT-PCR were designed using prime 5 software, as shown in Appendix A.

### 4.6. Measurements of Endogenous Plant Hormones

The contents of endogenous plant hormones were measured according to the study previously reported by Qi et al. [55]. Tomato leaves (about 1 g) were ground with a mortar and pestle in 3 mL of cooled 80% (*v*/*v*) methanol solution. After incubation for 12 h at 4 °C, the extracts were collected after centrifugation for 15 min under refrigerated conditions at 4 °C. The residues were suspended in 2.5 mL of extraction solution and stored at 4 °C for 1 h, then centrifuged again at 3059× *g* for 15 min at 4 °C. The extraction was repeated twice, the supernatant was merged, and the volume was increased to 10 mL with 80% pure methanol. Two milliliters of the supernatant were evaporated in a rotary vacuum at 38 °C for 4 h and then dissolved with 2 mL of 80% cold pure methanol. Finally, to establish liquid chromatograph detection, the extract was filtered with a 0.22 μm filter. The contents of IAA and ABA were detected by Quaternary gradient ultra-fast liquid chromatograph using Waters Acquity ARC 600-2998 equipped with the Symmetry-C18 column (4.6 × 250 mm, 5 μm). The SA content was measured by Liquid chromatography–mass spectrometry equipped with a C-18 column (2.1 × 50 mm, 1.8 μm, Agilent, Agilent Technologies Co. Ltd., Santa Clara, CA, USA) at a flow rate of 0.3 mL min^−1^.

ETH production was measured according to the method of Wang et al. [56] with minor modifications. The four fully expanded upper leaves beneath the growing tips of plants with different treatments were sampled and then placed in a 795 mL desiccative airtight container and incubated at 25 °C for 12 h. A quantity of 1 mL of headspace gas from each container was collected using a gas-tight hypodermic syringe and immediately injected into a gas chromatograph (GC-17A, Shimadzu, Kyoto, Japan) equipped with a flame ionization detector and an activated alumina column to determine the ethylene concentration.

### 4.7. Measurements of Photosynthetic Pigment Contents and Chlorophyll Fluorescence Parameters

The chlorophyll fluorescence parameters of tomato leaves were measured using an FMS-2 pulse-modulated fluorometer (Hansatech Instruments Ltd., Norfolk, UK). The FS under light adaptation, maximum photochemical efficiency (Fv/Fm), actual photochemical efficiency (φPS II), non-photochemical quenching (NPQ), and photochemical quenching (qP) were obtained under dark adaptation [57]. The contents of photosynthetic pigments (chlorophyll a, chlorophyll b, and carotenoids) in fresh tomato leaves were measured [22], and each sample (1 g) was transferred to a 20 mL tube with 10 mL of 80% acetone and then placed in the dark for 48 h (shaken every 8 h). When the leaves turned white, the tube was fixed with 80% acetone to 25 mL, and the absorbance was measured at 665, 649, and 470 nm.

### 4.8. Statistical Analysis

Each treatment contained 3 replicates, with each replicate consisting of 5 tomato seedlings. The experiment was completely randomized. The data were analyzed using SPSS 22.0 (SPSS Inc., Chicago, IL, USA). The values were expressed as the means ± SE of three independent experiments with 3 replicated measurements. The different treatments were analyzed by Duncan’s multiple range test at a *p* < 0.05 level.

## 5. Conclusions

In conclusion, this study clarifies the vital role of NO in alleviating salt stress in tomato seedlings. Moreover, the transcriptome analysis identified a large number of candidate genes involved in the alleviation of salt stress through pairwise comparisons. The results indicate a mechanism of NO-relieved of salt toxicity that involves two pathways: the signal transduction and synthesis of plant hormones, especially IAA, ETH, ABA, and SA, and photosynthesis. Collectively, these findings provide new insights into the biological functions of NO in plant growth and stress response. However, the mechanisms underlying NO and abiotic stress are quite complex, and future works should be established to characterize the functional roles of these DEGs.

## Figures and Tables

**Figure 1 ijms-23-04539-f001:**
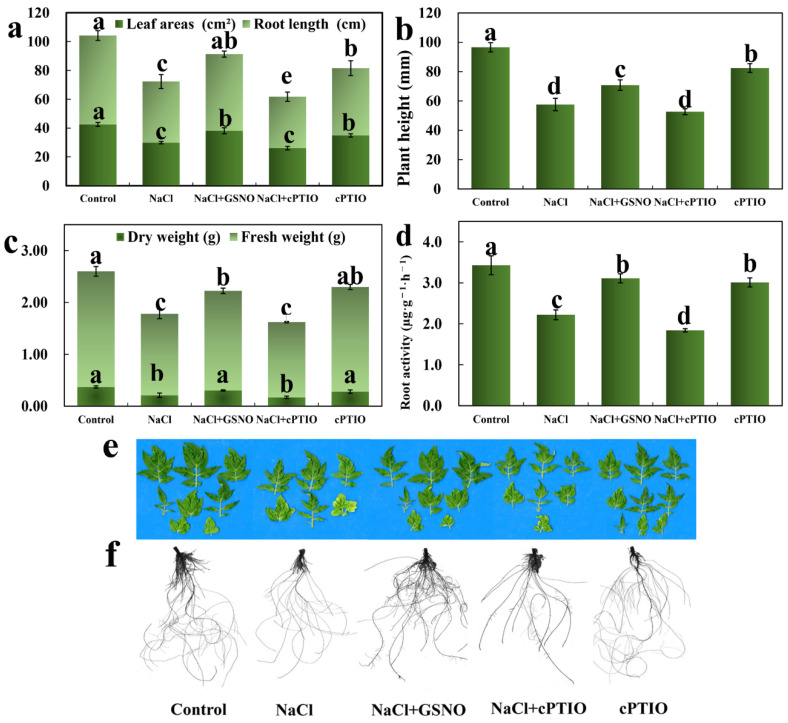
Effects of GSNO on phenotypic and physiological indexes of tomato seedlings during salt stress. (**a**) Leaf area and total root length. (**b**) Plant height. (**c**) Dry weight and fresh weight of the aboveground part of the plant. (**d**) Root activity. (**e**,**f**) Photographs were taken after 7 days of different treatments. The values (means ± SE) are the averages of three independent experiments (*n* = 15). Different letters mean significant difference by Duncan’s multiple range test (*p* < 0.05).

**Figure 2 ijms-23-04539-f002:**
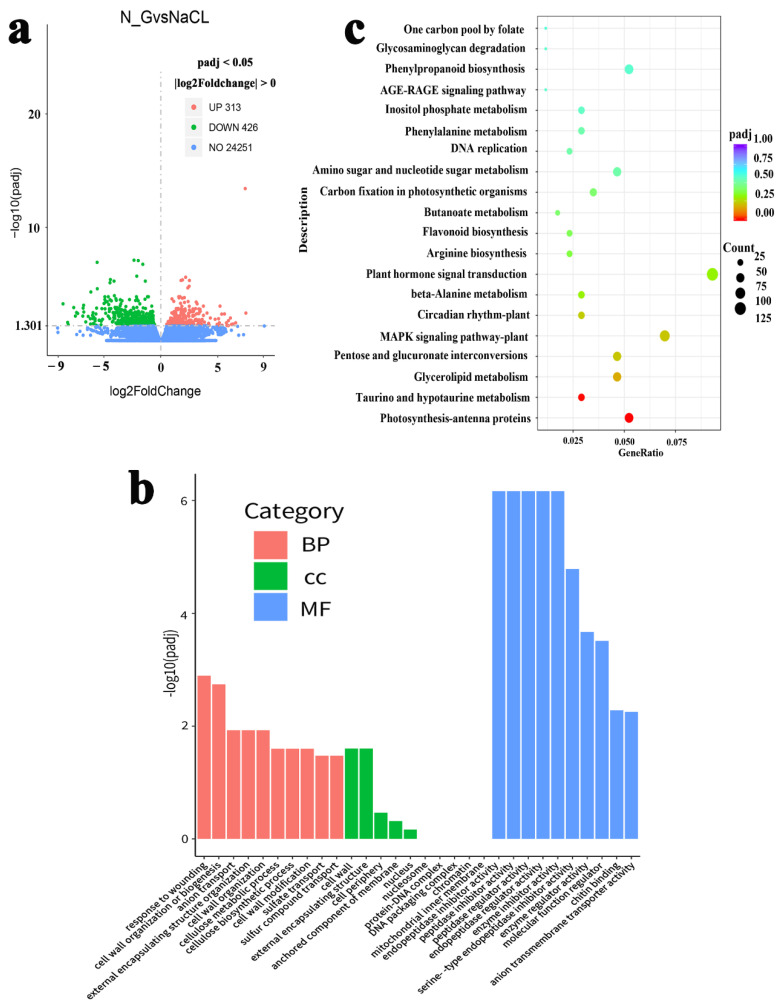
Transcriptome profiling analysis during NO alleviation of salt stress in tomato seedlings. (**a**) The number of DEGs is shown by the volcano plot. Red and green points represent up- and downregulated genes, respectively, and blue points represent no differences. (**b**) Functional gene ontology (GO) enrichment analysis of DEGs. GO terms are summarized in three categories: cellular component, molecular function, and biological process. (**c**) The DEG enrichment in different KEGG pathways. N_G: NaCl + GSNO; vs.: versus.

**Figure 3 ijms-23-04539-f003:**
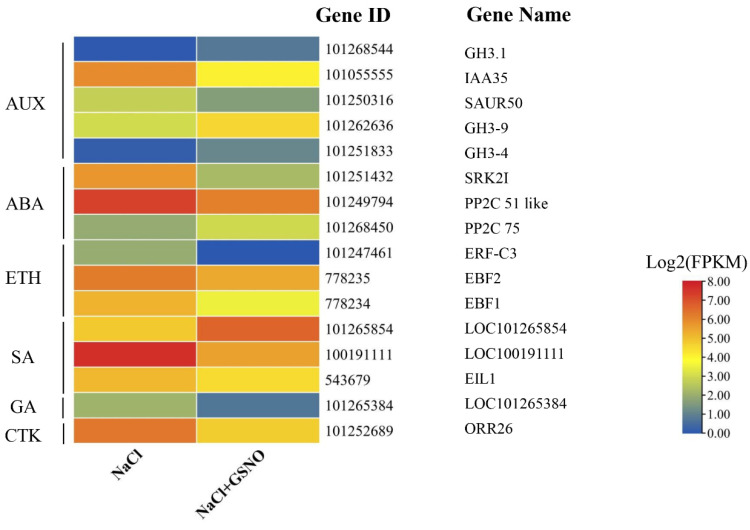
The heatmap of the expression of DEGs related to plant hormone signal transduction in NaCl and NaCl + GSNO treatments. The color scale corresponds to log 2-transformed (fragments per kilobyte per million reads) FPKM values. Red represents upregulation and blue represents downregulation of DEG expression.

**Figure 4 ijms-23-04539-f004:**
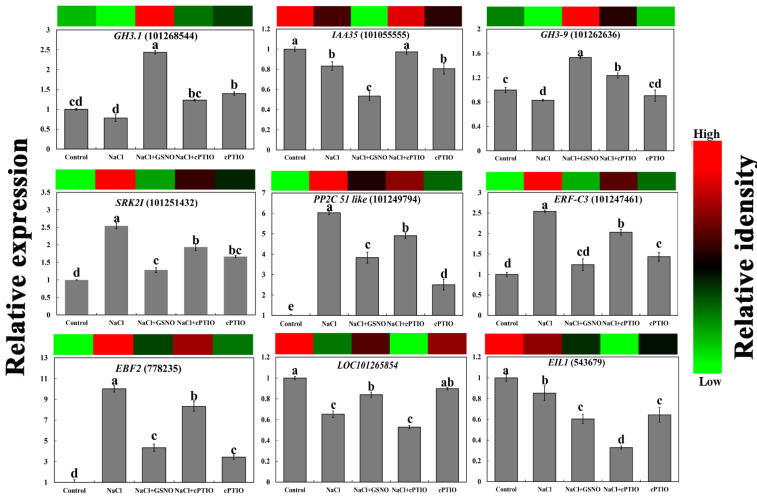
Quantitative real-time PCR analysis of the expression profiles of nine genes related to hormone signal transduction during NO alleviation of salt stress. The color scale corresponds to relative expression values of genes, with red indicating upregulation and green indicating downregulation. The error line is the standard error. Different letters mean significant difference by Duncan’s multiple range test (*p* < 0.05).

**Figure 5 ijms-23-04539-f005:**
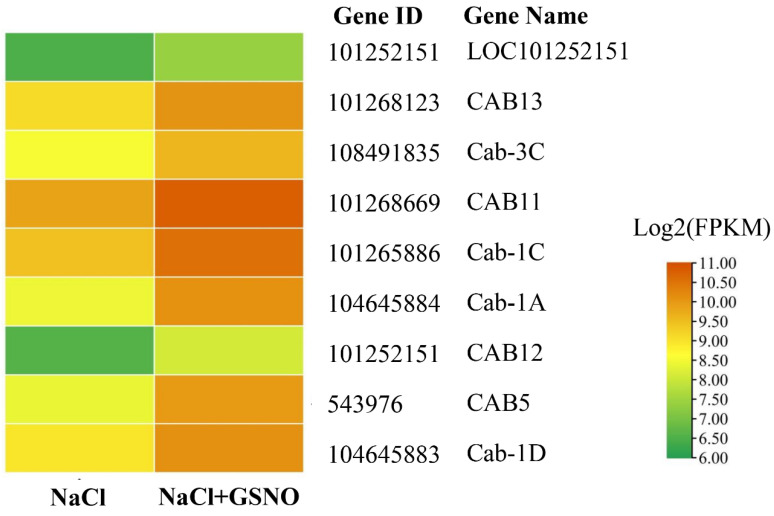
Expression patterns of differentially expressed genes related to photosynthesis in NO alleviation of salt stress. The color scale corresponds to log 2-transformed (fragments per kilobyte per million reads) FPKM values, with red indicating upregulation and green indicating downregulation. Each row represents a unigene. NaCl and NaCl + GSNO represent different treatments.

**Figure 6 ijms-23-04539-f006:**
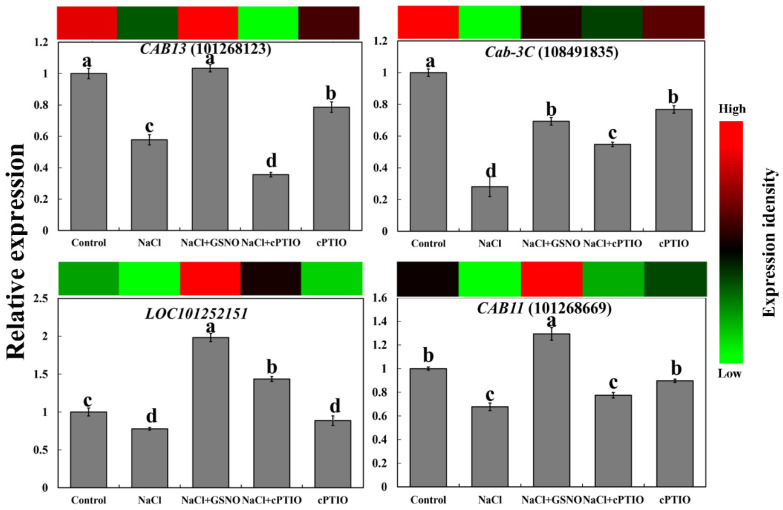
Quantitative real-time PCR analysis of the expression profiles of four genes related to photosynthesis during NO alleviation of salt stress. The color scale corresponds to relative expression values of genes, with red indicating upregulation and green indicating downregulation. The error line is the standard error. Different letters mean significant difference by Duncan’s multiple range test (*p* < 0.05).

**Figure 7 ijms-23-04539-f007:**
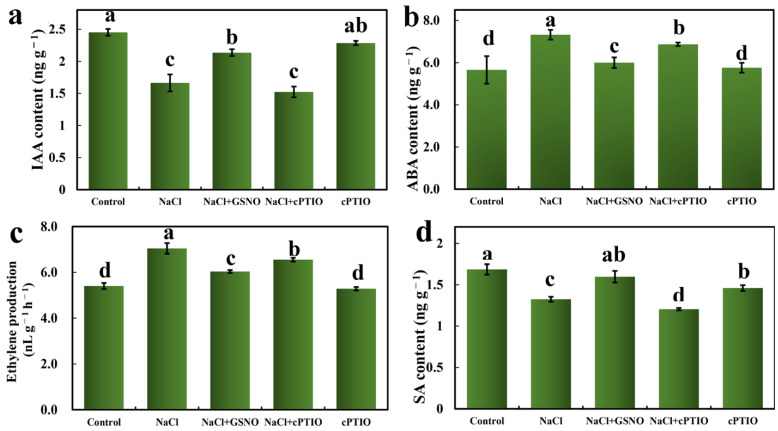
Changes in IAA, ABA, SA, and ethylene content during NO alleviation of salt stress. (**a**) IAA content, (**b**) ABA content, (**c**) ethylene content, (**d**) SA content. The values (means ± SE) are the averages of three independent experiments (*n* = 15). Bars not sharing the same letters indicate statistically significant differences by Duncan’s multiple range test (*p* < 0.05).

**Figure 8 ijms-23-04539-f008:**
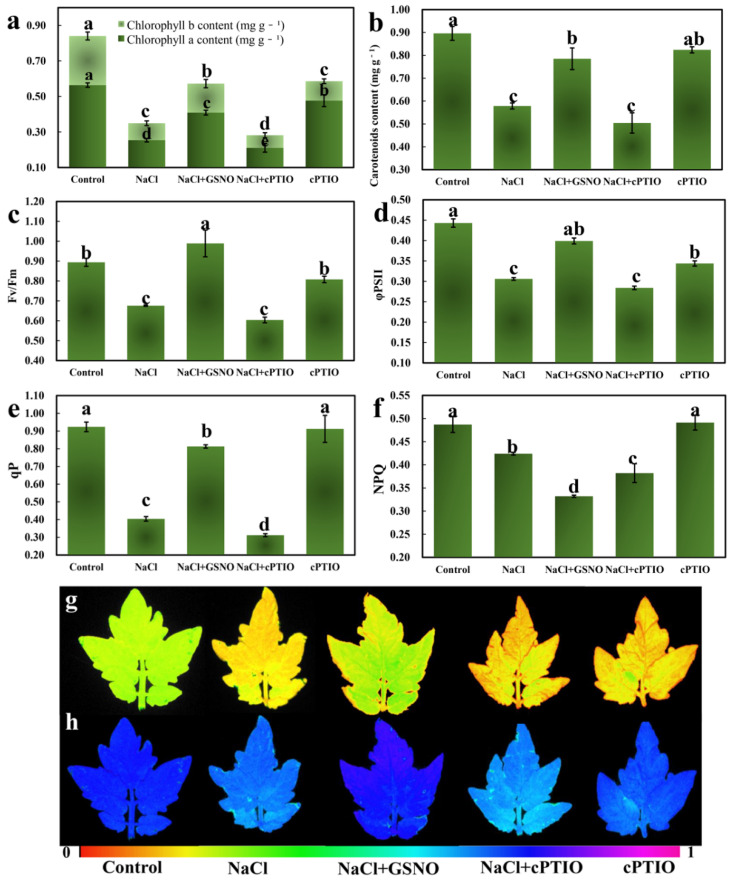
Changes in photosynthesis pigments during NO alleviation of salt stress. (**a**) Chlorophyll content, (**b**) carotenoid content, (**c**) maximum photochemical efficiency of PSII, (**d**) the quantum efficiency of PSII photochemical center, (**e**) the photochemical quenching coefficient, and (**f**) the non-photochemical quenching coefficient. (**g**,**h**) Images of Fv/Fm and ΦPSII, respectively. The values (means ± SE) are the averages of three independent experiments (*n* = 15). Bars not sharing the same letters indicate statistically significant differences by Duncan’s multiple range test (*p* < 0.05).

## Data Availability

Data are contained within the article or Appendix A.

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
