# Peer review of "Nitric Oxide Enhanced Salt Stress Tolerance in Tomato Seedlings, Involving Phytohormone Equilibrium and Photosynthesis"

_ijms, 2022, doi:10.3390/ijms23094539_

Round 1

Reviewer 1 Report

Results

I suggest the authors to remove some sentences form the results and write them in the discussion section:

  1. page 3. “Thus, these results indicated that NO is responsible for enhancing the tolerance of tomato seedlings to salt stress.”
  2. page 5. “implying that NO might mediate degradation of Aux/IAA repressors to induce auxin-regulated responses.”
  3. page 6. “Interestingly, GSNO-induced the regulation on these genes expression under stress was reversed by adding NO scavenger cPTIO (Figure 4), further conforming the necessary roles of NO in regulating the plant hormone signal transduction under salt stress.”
  4. page 7. “indicating the important roles of NO in enhancing photosystem II capacity under salt stress.”
  5. page 7. “These results indicate that NO might alleviate salt stress in tomato seedlings by activating photosystem.”

Legend of Figure 1 (page 3). Please write that “Dry weight and fresh weigh of the aboveground part of the plant”.

Figure 4 (page 7). I encourage the authors to use letters to indicate the statistically significant differences between the treatments.

Figure 6 (page 9). I encourage the authors to use letters to indicate the statistically significant differences between the treatments.

Material and methods

 “4.2. Measurement of morphological indexes” (page 14). We always measure the dry weight after drying the plant tissues until constant weight.

“4.3. Measurement of physiological indexes” (page 14). I suggest the authors to write the modifications for the method they used in order to measure the root activity.

“4.8. Statistical analysis” (page 16). Please define the number of replicates per experiment and the number of plants per replicate. In some figures (e.g. 8) the authors write n=15. Do the authors mean that they used five plants per treatment per experiment? Did the authors use only one plant per treatment per experiment (i.e. three plants per treatment)? In such a case, the number of plants is very small for some measurements (e.g. root length, leaf area, dry weight, fresh weight, hormone levels) in agricultural experimentation, even under controlled conditions.

References

  1. Please use italics for the journal volume (see Ref. 1, 36).
  2. Please use italics for the scientific names (see Ref. 47, 49, 54).
  3. Please write the journal volume (Ref. 4, 10).
  4. The references from books are not written according to the instructions (see Ref. 19, 35, 45).
  5. Please use correct abbreviations for journal titles (see Ref. 9, 26-dot after Environ, 49, 55).
  6. Please correct the pages (see Ref. 10, 46).

Author Response

Response to Reviewer #1

Comment 1:

Results

I suggest the authors to remove some sentences form the results and write them in the discussion section:

  1. page 3. “Thus, these results indicated that NO is responsible for enhancing the tolerance of tomato seedlings to salt stress.”
  2. page 5. “implying that NO might mediate degradation of Aux/IAA repressors to induce auxin-regulated responses.”
  3. page 6. “Interestingly, GSNO-induced the regulation on these genes expression under stress was reversed by adding NO scavenger cPTIO (Figure 4), further conforming the necessary roles of NO in regulating the plant hormone signal transduction under salt stress.”
  4. page 7. “indicating the important roles of NO in enhancing photosystem II capacity under salt stress.”
  5. page 7. “These results indicate that NO might alleviate salt stress in tomato seedlings by activating photosystem.”

Legend of Figure 1 (page 3). Please write that “Dry weight and fresh weigh of the aboveground part of the plant”.

Figure 4 (page 7). I encourage the authors to use letters to indicate the statistically significant differences between the treatments.

Figure 6 (page 9). I encourage the authors to use letters to indicate the statistically significant differences between the treatments.

Response:

We would like to thank the reviewer for the thoughtful comments and constructive suggestions, which help to improve the quality of this manuscript.

  • According to your suggestions, we have removed these sentences you mentioned above and wrote them in the discussion section (More details please see the revised manuscript).
  • According to your suggestion, we have added “Dry weight and fresh weigh of the aboveground part of the plant” in legend of Figure 1 of revised manuscript.
  • We have added letters in Figures 4 and 6 to indicate the statistically significant differences between different treatments (More details please see the revised Figures).

Comment 2:

Material and methods

 “4.2. Measurement of morphological indexes” (page 14). We always measure the dry weight after drying the plant tissues until constant weight.

“4.3. Measurement of physiological indexes” (page 14). I suggest the authors to write the modifications for the method they used in order to measure the root activity.

“4.8. Statistical analysis” (page 16). Please define the number of replicates per experiment and the number of plants per replicate. In some figures (e.g. 8) the authors write n=15. Do the authors mean that they used five plants per treatment per experiment? Did the authors use only one plant per treatment per experiment (i.e. three plants per treatment)? In such a case, the number of plants is very small for some measurements (e.g. root length, leaf area, dry weight, fresh weight, hormone levels) in agricultural experimentation, even under controlled conditions.

Response:

  • Thank you very much for pointing out this problem, in our preliminary experiment, after drying of the aboveground part of the plant for 48 hours, dry weight is until constant weight. Thus, in order to describe the method in more detail and objectively, we have changed related statement “After the plant samples were dried at 80℃ for 48 h, which was until constant weight, the DW was measured” in part of 4.2.
  • Thank you very much! we have added the modifications for the method about root activity in part of 4.3. The related statements are as follows:

The root activity was measured by the triphenyl tetrazolium chloride (TTC) method as described by Yan et al. [32] with minor modification. The tomato roots (about 1 g) were washed with deionized water and excised at 2 cm from the root tips on 2 h with a mixture of 7 mL 0.4% (w/v) TTC and 7 mL phosphate buffer (pH 7.5) at 37 â—¦C, and then added 4 mL of H2SO4 (1 M). Roots were dried with filter paper and extracted by ethyl acetate. The absorbance was measured at 485 nm.

  • Thank you! According to your comment, we have defined the number of replicates per experiment and the number of plants per replicate in revised manuscript in part of 4.8. In our study, each treatment contained 3 replicates, with each replicate consisting of 5 tomato seedlings.

Comment 3:

References

  1. Please use italics for the journal volume (see Ref. 1, 36).
  2. Please use italics for the scientific names (see Ref. 47, 49, 54).
  3. Please write the journal volume (Ref. 4, 10).
  4. The references from books are not written according to the instructions (see Ref. 19, 35, 45).
  5. Please use correct abbreviations for journal titles (see Ref. 9, 26-dot after Environ, 49, 55).
  6. Please correct the pages (see Ref. 10, 46).

Response:

Thank you very much for your comments, and we are very sorry for these mistakes. First, according to your suggestions, we have revised the format of these references you mentioned above in revision. Besides, we have also carefully checked and revised all references format by respecting the Guide for authors (More details please see the revised references in revision) 

Reviewer 2 Report

This study clarifies a vital role of Nitric oxide in alleviating salt stress in tomato seedling.

The study is interesting and the methods are adequately described.

Minor comments:
- You should respect the Guide for authors (style and text format).
- Authors should enlarge figure 2.
- Avoid using the color black in histograms. 

Author Response

Response to Reviewer #2

Comment:

Minor comments:
- You should respect the Guide for authors (style and text format).
- Authors should enlarge figure 2.
- Avoid using the color black in histograms

Response:

Thank you for your valuable suggestion, which help to improve the quality of this manuscript.

  • We have carefully revised style and text format according to the Guide for authors in the revised manuscript (More details please see the revised manuscript).
  • According to your suggestion, we have enlarged figure 2 to make it clear to the readers (More details please see the revised figure 2 in revision).
  • According to your comment, we have revised all black histograms by using the color green in histograms (More details please see the revised figures in revision).
